# Genomic Landscape of Angiosarcoma: A Targeted and Immunotherapy Biomarker Analysis

**DOI:** 10.3390/cancers13194816

**Published:** 2021-09-26

**Authors:** Andrea Espejo Freire, Andrew Elliott, Andrew Rosenberg, Philippos Apolinario Costa, Priscila Barreto-Coelho, Emily Jonczak, Gina D’Amato, Ty Subhawong, Junaid Arshad, Julio A. Diaz-Perez, Wolfgang M. Korn, Matthew J. Oberley, Daniel Magee, Don Dizon, Margaret von Mehren, Moh’d M. Khushman, Atif Mahmoud Hussein, Kirsten Leu, Jonathan C. Trent

**Affiliations:** 1Department of Medicine, Hematology & Oncology, Sylvester Comprehensive Cancer Center, Jackson Memorial Hospital, Miller School of Medicine, University of Miami, Miami, FL 33136, USA; andrea.espejofreire@jhsmiami.org (A.E.F.); philippos.costa@jhsmiami.org (P.A.C.); priscila.barretocoe@jhsmiami.org (P.B.-C.); eej18@med.miami.edu (E.J.); gina.damato@med.miami.edu (G.D.); 2Department of Clinical and Translational Research, Caris Life Sciences, Phoenix, AZ 85040, USA; aelliott@carisls.com; 3Department of Pathology, Sylvester Comprehensive Cancer Center, Jackson Memorial Hospital, Miller School of Medicine, University of Miami, Miami, FL 33136, USA; arosenberg@med.miami.edu (A.R.); julio.diazperez@jhsmiami.org (J.A.D.-P.); 4Department of Radiology, Sylvester Comprehensive Cancer Center, Jackson Memorial Hospital, Miller School of Medicine, University of Miami, Miami, FL 33136, USA; tsubhawong@med.miami.edu; 5Department of Medicine, Medical Oncology, The University of Arizona College of Medicine, University of Arizona Cancer Center, Tucson, AZ 85724, USA; junaidarshad@email.arizona.edu; 6Department of Medical Affairs, Caris Life Sciences, Phoenix, AZ 85040, USA; wmkorn@carisls.com; 7Department of Pathology and Genetics, Caris Life Sciences, Phoenix, AZ 85040, USA; moberley@carisls.com; 8Department of Cognitive Computing, Caris Life Sciences, Phoenix, AZ 85040, USA; dmagee@carisls.com; 9Department of Medical Oncology and Gynecologic Medical Oncology, Lifespan Cancer Institute, Rode Island Hospital, Providence, RI 02903, USA; don.dizon@lifespan.org; 10Department of Hematology & Oncology, Fox Chase Cancer Center, Temple Health, Philadelphia, PA 19111, USA; margaret.vonmehren@fccc.edu; 11O’Neal Comprehensive Cancer Center, Department of Medicine, Hematology & Oncology, The University of Alabama at Birmingham, Birmingham, AL 35233, USA; mkhushman@uabmc.edu; 12Department of Hematology & Oncology, Memorial Health Care System, Memorial Cancer Institute, Hollywood, FL 33021, USA; ahussein@mhs.net; 13Medical Oncology, Nebraska Cancer Specialists, Omaha, NE 68114, USA; kleu@nebraskacancer.com

**Keywords:** Angiosarcoma, biomarkers, tumor microenvironment, immunotherapy, next-generation sequencing, whole transcriptome sequencing

## Abstract

**Simple Summary:**

Angiosarcomas (AS) are rare, highly aggressive sarcomas with limited therapeutic options. Genomic sequencing techniques have identified recurrent genetic abnormalities. Nevertheless, the association of these findings with etiology, site of origin, prognosis, and therapeutic implications is not well understood. We analyzed Next Generation Sequencing (NGS) and Whole Transcriptome Sequencing (WTS) data in a cohort of 143 AS cases. We identified distinct genomic biology according to the AS primary site. Head and neck AS cases primarily have Immunotherapy (IO) response markers and mutations in *TP53* and *POT1*. On the other hand, breast AS is enriched for cell cycle alterations, predominately *MYC* amplification. Additionally, a microenvironment with abundant immune cells is present in a minority of cases but distributed evenly among primary sites. Our findings can facilitate the design and optimization of therapeutic strategies for AS according to its biology at different primary sites.

**Abstract:**

We performed a retrospective analysis of angiosarcoma (AS) genomic biomarkers and their associations with the site of origin in a cohort of 143 cases. Primary sites were head and neck (31%), breast (22%), extremity (11%), viscera (20%), skin at other locations (8%), and unknown (9%). All cases had Next Generation Sequencing (NGS) data with a 592 gene panel, and 53 cases had Whole Exome Sequencing (WES) data, which we used to study the microenvironment phenotype. The immunotherapy (IO) response biomarkers Tumor Mutation Burden (TMB), Microsatellite Instability (MSI), and PD-L1 status were the most frequently encountered alteration, present in 36.4% of the cohort and 65% of head and neck AS (H/N-AS) (*p* < 0.0001). In H/N-AS, TMB-High was seen in 63.4% of cases (*p* < 0.0001) and PDL-1 positivity in 33% of cases. The most common genetic alterations were *TP53* (29%), *MYC* amplification (23%), *ARID1A* (17%), *POT1* (16%), and *ATRX* (13%). H/N-AS cases had predominantly mutations in *TP53* (50.0%, *p* = 0.0004), *POT1* (40.5%, *p* < 0.0001), and *ARID1A* (33.3%, *p* = 0.5875). In breast AS, leading alterations were *MYC* amplification (63.3%, *p* < 0.0001), *HRAS* (16.1%, *p* = 0.0377), and *PIK3CA* (16.1%, *p* = 0.2352). At other sites, conclusions are difficult to generate due to the small number of cases. A microenvironment with a high immune signature, previously associated with IO response, was evenly distributed in 13% of the cases at different primary sites. Our findings can facilitate the design and optimization of therapeutic strategies for AS.

## 1. Introduction

Angiosarcomas (AS) are highly aggressive sarcomas that account for only 2% of all soft-tissue-sarcomas (STS) [1]. Unfortunately, even when patients present with localized disease, over 50% will relapse after initial treatment, resulting in a five-year OS of only 60%. Furthermore, once patients have locally advanced or metastatic disease, the median OS is only 9–15 month [2,3,4]. Cytotoxic chemotherapy frequently shows activity, but tumor responses are short-lived, and most patients ultimately die from metastatic disease [3,4]. Moreover, despite evidence of upregulation of vascular-specific receptor tyrosine kinases, VEGF blockade provides at most a 2–4-month survival benefit [5,6,7,8,9]. Lately, growing evidence of immunotherapy (IO) activity in AS has emerged [10,11]. However, not all AS primary sites show uniform responses, and ultimately IO’s role in the treatment of AS is not clearly defined.

At the different sites of origin, cases of AS show different clinical features and prognosis. The most common AS location is head and neck AS (H/N-AS), followed by breast AS (B-AS), visceral, other cutaneous sites, and the extremities. The majority of cases of AS occur sporadically (primary) or are related to radiation therapy or chronic lymphedema (secondary) [12]. A French retrospective multicenter study of 161 patients reported that visceral (heart, liver, and spleen) and primary bone sites were associated with worse prognosis [13]. In a study of 200 AS cases from China that also showed biological differences, the worst prognosis was seen in H/N-AS (5-year OS of 28%), followed by visceral (37%), and B-AS (87%) [14]. Evidence shows that patients with secondary B-AS have a more aggressive tumor phenotype and worse survival outcome than patients with primary B-AS [14,15]. A study of over 470 patients extracted from the SEER database described that secondary B-AS appears in older patients and presents with more locally advanced stage (57% vs. 18%) and high grade (58% vs. 32%). In this cohort, the median OS was 93 months for primary B-AS and 32 months for secondary B-AS [15].

Along with the differences in clinical behavior, some small cohorts in the literature show genomic differences within AS. The first identified genetic alteration was *KDR* (AKA *VEGFR2*), which harbors point mutations in 10% of primary or secondary B-AS [16]. Other recurrent alterations are *TP53, PIK3CA, POT1, RAS, BRAF, PTPRB, PLCG1*, and *APC* [2,11,17]. Some mutations appear to be distinct to cases of primary and secondary AS. *MYC* amplification was reported in 50 to 100% of radiation-associated AS cases but not in primary AS [18,19,20]. Most recently, Whole Exome Sequencing (WES) results of 47 samples from 36 patients self-registered to the Angiosarcoma Project were published. In this cohort, the authors reported that *TP53* and *KDR* mutations are mutually exclusive, with 89% of *KDR* mutations in primary B-AS compared to 82% of TP53 in non-primary B-AS (*p* = 0.02). Nine out of ten *PIK3CA* alterations were also seen in primary B-AS (*p* = 0.0003) [11]. Despite sequencing techniques allowing the identification of recurrent somatic genetic abnormalities, the rarity of AS challenges our efforts to establish strong associations with the site of origin, etiology, and therapeutic implications.

There is growing evidence that IO is highly active for some patients with AS and that IO activity is likely dependent on the site of origin. First, a phase II study on the use of immunotherapy for advanced STS (Alliance A091401) showed that one AS patient had an objective response [21]. Subsequently, a retrospective analysis of seven AS cases treated with immunotherapy revealed a response rate of 71% (5/7) at 12 weeks, including one case of complete response [10]. Here H/N-AS cases were four out of five responders. Finally, in The Angiosarcoma Project, 3 out of 10 patients with H/N-AS received immunotherapy (IO), and two achieved exceptional responses. In contrast, none of the three patients with AS other than H/N treated with immunotherapy responded to the therapy [11].

As responses to IO are not homogeneous for specific histology, efforts to determine potential IO response markers are in progress. The Angiosarcoma Project identified that the median tumor mutation burden (TMB) was significantly higher in patients with H/N-AS (*p* = 1.10 × 10^−5^). In this cohort, both cases benefiting from IO had very high TMB, with 78 and 138 mutations/MB [11]. However, experience in IO for STS trials has taught us that classic IO response markers, TMB and PDL-1, are not the sole determinants of response. Transcriptomic analysis is now available to estimate the relative abundance of immune and stromal cells within tumor samples. Using this technique, Petiprez et al. described a classification of STS based on their tumor microenvironment. In the SARC028 trial for the use of PDL-1 blockage for STS, they identified that an immune-rich microenvironment, particularly a B cell abundance, correlated with better response rate and improved PFS [22]. Interestingly, the overall TMB appeared similar across all classes of microenvironment phenotypes. In other histologies, microenvironment analysis also shows predictive capabilities for IO and other targeted therapies. For example, in renal cell carcinoma, gene expression signatures of angiogenesis, T-effector, and myeloid cells are predictive of PFS for IO alone or combined with anti-VEGF blockage [23]. Whether these methods can be applied similarly to patients with AS needs further investigation.

Here, we analyzed genomic data of Next Generation Sequencing (NGS) and Whole Transcriptome Sequencing (WTS) from 143 cases. To our knowledge, this is the largest cohort of AS cases with genomic data. In addition, we described a particular AS biology according to the primary site and showed potential biomarkers, including a description of the microenvironment to guide future therapeutic studies.

## 2. Materials and Methods

We retrospectively analyzed the data of 143 AS tumors profiled by Caris Life Sciences from 2015–2019. We included the annotations of “Angiosarcoma”, “Angiomyosarcoma”, or “Lymphangiosarcoma”. Clinical characteristics including age, sex, site of origin, site of biopsy, and the status of metastatic vs. primary were tabulated. No data on prior exposure to radiation therapy were available. NGS enriched for 592 cancer-related whole-gene targets was performed on each tumor. We included pathogenic mutations and copy number amplification in the analysis.

WTS was performed on 53 tumors and used for microenvironment cell population (MCP)-counter analysis, as described by Becht et al. [24]. First, we estimated a cell population of interest using transcriptomic markers (TMs). TMs are gene expression features expressed in one and only one cell population. The method generates an abundance score for CD3+ T cells, CD8+ T cells, cytotoxic lymphocytes, NK cells, B lymphocytes, monocyte lineage cells, endothelial cells, and fibroblasts [24]. Next, we identified subgroups based on tumor microenvironment profiles by hierarchical clustering of MCP-counter Z-scores [22].

Biomarkers classically associated with response to IO (TMB-High (≥10/Mb), MSI-High, and PD-L1 (IHC ≥2+ and 5%) were included. A sarcoma pathologist at Sylvester Comprehensive Cancer Center reviewed the hematoxylin and eosin (H&E) slides to confirm the diagnosis. Additionally, we annotated data of cell morphology, anatomical biopsy site, grade, necrosis, lumen formation, and intra and peritumoral inflammatory infiltrate. The inflammatory infiltrates were graded as follows: 0—no inflammatory cells observed, 1—corresponding to <5% of the cellularity, 2—corresponding to 5–30% of the cellularity, and 3—corresponding to >30% of the cellularity.

Cytologic, molecular, and genomic results were evaluated according to the primary tumor site. Statistical analyses were performed using Chi-square or Fisher’s exact tests, where appropriate. The Wilcoxon Method was used to compare groups, and *p*-values were adjusted for multiple hypothesis testing using the Benjamini and Hochberg procedure.

## 3. Results

The cohort’s median age was 67 (range 22–89); 61% were female and 29% were metastatic/recurrent. The number of cases by location were head and neck (*n* = 44, 31%), breast (*n* = 31, 22%), extremity (*n* = 16, 11%), viscera (*n* = 28, 20%), skin at other locations (*n* = 11, 8%), and unknown (*n* = 13, 9%). Table 1 shows the H&E histologic characteristics of the cases. Figure 1 shows the spectrum of the density of inflammation within cases of AS.

### 3.1. Markers of Immunotherapy Response

Predictive IO-response biomarkers were the most common marker in the entire cohort, present in 36.4% of cases (TMB-High in 26%, PD-L1+ 21.8%, MSI-High 0.7%). Predictive IO-response biomarkers were the highest in the H/N-AS subgroup, with TMB-High observed in 63.4% of H/N-AS cases (*n* = 26/41; *p* < 0.0001), a significant increase compared to other sites. Fourteen cases of H/N-AS (33%) were positive for PD-L1 by IHC, 11 of which were concurrently TMB-High. Only one case of H/N-AS had dMMR/MSI-high status. TMB-High was present in a few cases at other locations: four visceral AS cases and one case in breast, extremity, and other cutaneous site. Similarly, PDL-1 positivity is present in six cases of visceral AS, three cases of B-AS, two cases of extremity AS, and one case of other cutaneous AS. Of note, B-AS had the lowest frequency of IO-response biomarkers. Figure 2 shows IO-response biomarkers.

### 3.2. Genetic Alterations

The most common genetic alterations were *TP53* (29%), *MYC* amplification (23%), *ARID1A* (17%), *POT1* (16%), and *ATRX* (13%). Genetic alterations were distinct according to the primary site. In H/N-AS, *TP53* mutations were present in 48.8% (*n* = 21/43; *p* = 0.0002), *POT1* in 41.9% (*n* = 18/43; *p* < 0.0001), and *ARID1A* in 31.3% (*n* = 5/16; *p* = 0.7331). On the other hand, in B-AS, cell cycle pathway aberrations were common, with *MYC* amplification present in 63.3% (*n* = 19/30; *p* < 0.0001). Mutations in *HRAS* were present in 16.1% (*n* = 5/31; *p* = 0.0155) and *PIK3CA* in 16.1% (*n* = 5/31; *p* = 0.1489) of B-AS cases. Interestingly, *MYC* amplification was also seen in 45.5% (5/11) of cutaneous cases at other locations than H/N or breast, and 37.5% (6/16) of extremity AS cases, but not seen in H/N or visceral AS cases. *MYC* amplification has been described in radiation-associated AS of the breast [17,18,19]; future studies are needed to investigate further whether *MYC* amplification is an etiologic factor at other sites of secondary AS. Unfortunately, our cohort did not include annotation of whether the patient had a prior history of radiation exposure. Finally, some distinct alterations appeared more commonly at other cutaneous, visceral, and extremity locations; however, conclusions are difficult to obtain due to the small number of cases at these sites. Figure 3 shows the genetic alterations in AS.

### 3.3. Microenvironment Phenotype

Using the MCP-counter method, we defined four distinct immune classes based on microenvironment cell population abundance. Hierarchical clustering identified subgroups with distinct microenvironment profiles consistent with those described by Petitprez et al. [22]. Fifty-three cases with available WTS data were distributed as follows: Immune-High—B lineage high (13.2%), Vascularized—Endothelial cells high (24.5%), Immune-Desert (41.5%), and Heterogeneous—Moderate abundance (20.8%). Immune class signatures were evenly distributed among different primary sites. Interestingly, the Immune-High group had the lowest median TMB: 6 muts/MB (range 3–17). See Figure 4. 

Next, we compared the microenvironment of the AS cohort with a cohort of melanoma (*n* = 1255). Of all tumors in humans, melanomas have one of the highest burdens of somatic genetic alterations [25]. Moreover, particularly in cutaneous melanoma, an extensive CD8+ T-cell infiltration has been described and associated with better prognosis and response to IO [26]. Interestingly, the microenvironment of angiosarcoma has a similar immune profile to melanoma but with enrichment of endothelial and myeloid dendritic cells (Figure 4b). While the median abundance of CD8+ T cell and B cell populations was lower in AS than melanoma, the difference was not statistically significant. Neutrophils, NK cells, and monocytic lineage cells had a comparable abundance.

Finally, when analyzing the microenvironment according to the tabulated histologic characteristics observed by H&E, we observed that the cases with an inflammatory infiltrate of grade 2 or 3 had a higher number of T cells, CD8+ cells, cytotoxic T cells, NK, and B cells using the MCP counter method. Importantly, B-cell abundance in cases with grade 2 or 3 infiltrate was significantly higher than in cases of grade 0 or 1 (*p* = 0.034). In addition, expression of immune-related markers TIM3, LAG3, PD-1 (PDCD1), PD-L1 (CD274), and PD-L2 (PDCD1LG2) was also more abundant in the cases with grade 2 or 3 inflammatory infiltrate. For PDL-1, this was statistically significant (*p* = 0.038). See Figure 5. 

## 4. Discussion

To our knowledge, this is the largest cohort of AS genomic biology described to date.

Our findings confirm previous studies that show that AS has distinct biology depending on its primary site and etiology. Here, we confirm that classical IO-response markers are common in AS, present in about one-third of the cases. We found that over 60% of cases of H/N-AS had markers of IO-response. Our findings corroborate the previously described cohorts where high TMB was clustered in 50–60% of cases of H/N-AS [11,27,28]. In these prior reports, UV light mutational signatures were described in cases of H/N-AS through whole genome and whole exome sequencing analysis. Unfortunately, our data comes from a specific 592 gene panel that involves sequencing a significantly smaller part of the genome. As such, we have limited power to present the results of mutational signatures. However, the consistency of other findings suggests that a UV mutational signature is likely to be found in cases of H/N-AS associated with high TMB. These factors may explain why H/N-AS cases benefit more from IO. The retrospective series by Florou et al. showed that most responders were cases of H/N-AS (four out of five cases) [10]. Similarly, the two cases of outstanding responses noted in the Angiosarcoma Project had TMB High H/N-AS [11].

In our cohort, TMB-High was also present in a few cases at other locations (in four cases of visceral AS and in one case of B-AS, extremity, cutaneous). PDL-1 positivity was present in 14 cases of H/N-AS, six cases of visceral AS, three cases of B-AS, two cases of extremity AS, and one case of cutaneous AS. Interestingly, in the cohort described by Florou et al., the case that achieved a long-lasting complete response had low TMB with only 0.9 muts/MB, and a patient with RT-associated B-AS also had a PR. These findings should indicate that IO-response markers are not the sole determinant of IO response. The opportunity for responses is not only seen at the H-N location.

A microenvironment with a high immune signature and abundance of B-cells was present in about 13% of the cases and evenly distributed among different primary sites. A signature of B-cell lineage abundance, regardless of high or low CD8+ T cell infiltration, appears to predict response to PD1 blockade and PFS in STS [22]. Interestingly, in this immune-high group, overall, the cases had a low TMB. The contribution of each of these factors and the dynamic microenvironment changes to IO response are still unknown. In other solid tumors, distinct microenvironment characteristics show predictor capabilities of IO and other targeted therapies [23,29]. However, not a single phenotype across solid tumors has yielded similar prognostic and predictive capabilities. Further studies are warranted to determine if a phenotype with an abundance of B cells in AS results in similar predictive capabilities to what has been seen for other STS. Additionally, the dynamics of the microenvironment upon treatment could potentially have better predictive capabilities. In melanoma, a highly immunogenic tumor where IO is active, INF-gamma driven infiltration of CD8+ lymphocytes upon treatment predicts responses [26,29]. In our cohort, we described a similar microenvironment to that of melanoma. Future studies should further examine the predictive capabilities of microenvironment analysis in AS to move forward to incorporating this method into clinical practice. WTS is commonly performed when profiling solid tumors; however, immune cell abundance and microenvironment cell counter results are not routinely reported to treating physicians. As more evidence of the predictive capabilities of this method emerges, we may be able to incorporate this information into clinical decision making as we move forward in tailoring effective therapies for rare tumors. Importantly, we reviewed the H&E slides of 138 of the patients (four cases had no available H&E slide) and described the tumor inflammatory infiltrate. We saw that B-cell abundance by WTS and PDL-1 expression was associated with the presence of an inflammatory infiltrate of grade 2 or 3, as assessed by light microscopy. Thus, we could optimize this strategy by incorporating immunohistochemistry to compare the predictive phenotypes determined by the MCP-counter method.

Certain genetic alterations of AS are more common at specific primary sites. Consequently, further studies to overcome IO resistance and increase the effectiveness of targeted therapies accounting for specific alterations are warranted. Therapeutic strategies that target *TP53, POT1*, and *ARID1A* could be of value in H/N-AS. *TP53* tumor suppressor activity triggers cell cycle arrest, death by apoptosis, and senescence by regulation of multiple pathways. Even though *TP53* is widely mutated in cancer, targeting it has been challenging. Recently, the study of small molecules to reestablish the activity of mutant p53 has shown promising results. In particular, APR-246 (eprenetapopt), which refolds mutant p53 to induce p53 target genes, demonstrated clinical activity in myeloid malignancies [30,31]. This promising strategy is currently under investigation in combination with IO, chemotherapy, and other targeted agents. In cases with a *TP53* mutation, we found that 67% had concomitant markers of IO response. Therefore, it may be essential to investigate if the use of this molecule could overcome IO resistance in H/N-AS. *ARID1A* is one of the most common alterations encountered in our cohort and is also seen predominately in H/N-AS. Being part of the chromatin remodeling complex SWI/SNF, its deficiency results in EZH2 overactivity. In epithelioid sarcoma, we observed success using EZH2 inhibitor tazemetostat [32,33,34]. This strategy has shown efficacy in *ARID1A* mutated ovarian and endometrioid cell carcinomas [35]. Therefore, prospective evaluation of tazemetostat in ARID1A mutated AS should be considered [36]. Lastly, *POT1* is involved in telomere maintenance, and its regular activity results in cell aging and apoptosis. Three percent of malignancies have *POT1* mutations; however, its prevalence is higher in AS (23%) and is described among the top predisposition genes for familial melanoma and cardiac AS [37]. In our cohort, it was predominately present in cases of H/N-AS. Unfortunately, the therapeutic role of *POT1* inhibition in cancer is currently unknown.

The molecular alterations in B-AS give additional opportunities for other therapeutic strategies. Over 60% of cases of B-AS show *MYC* amplification. In our cohort, we do not have data about the association of the cases with prior radiation therapy. However, strong evidence shows that *MYC* amplification is almost exclusively seen in radiation-associated AS [18,20]. MYC proteins coordinate transcription, DNA replication, and cell cycle progression. Strategies to target cell cycle by CDK inhibition have shown promising results in *MYC* amplified tumors. In neuroblastoma cell lines, CDK/CDK1 inhibitors show an ability to downregulate *MYC* [38]. Recently, fadraciclib, a potent inhibitor of CDK9 showed an ability to repress *MYC* and is currently in early-phase clinical trials for solid tumors and hematologic malignancies [39,40]. Additionally, bromodomain inhibitors (BET) have proven efficacy in *MYC* amplified lymphoma. Their use in combination with CDK2 inhibitor is being studied in *MYC*-driven medulloblastoma [41,42,43]. Finally, in ovarian cancer, *MYC* amplification predicted synergistic benefit from a combination of PARP inhibition with olaparib and CDK-4 inhibition with palbociclib [44]. These strategies deserve further study for *MYC* amplified AS. Other common mutations found in B-AS are *PIK3CA* and *HRAS*. Prior reports indicate that *PIK3CA* mutations are primarily found in cases of primary B-AS. PI3K inhibition is efficacious in *PIK3CA* mutated breast cancer. These mutations found in primary breast AS suggest that site of origin may predispose to overactive *PIK3CA*-driven AS. Here, evaluation of the use of alpelisib or other PI3K inhibitors is necessary for cases of primary breast AS [45]. These differences should be accounted for when devising differential strategies to treat RT-associated B-AS and primary B-AS. On the other hand, mutations in *RAS* usually result in downstream activation of MAPK and PI3K. Several compounds targeting downstream effects of RAS, such as MEK, AKT, and PI3K are available and should be considered in these cases [46]. This agent’s synergistic effects in combination with IO are also being studied [46,47].

Finally, we acknowledge that an important constraint of our study is the limited clinical data obtained from the requisition forms submitted by the ordering physician. Because preclinical models of AS are limited, this type of genomic analysis should be performed in retrospective and prospective cohorts, with available clinical data of responses to the different therapeutic strategies to understand the therapeutic implications further. Importantly, a recent study showed similarities of the AS genomic landscape to that of canine hemangiosarcoma. In canine hemangiosarcoma, the recurrent oncogenic mutations were *TP53* (66%), *PIK3CA* (46%), *NRAS* (24%), *PTEN* (6%), and *PLCG1* (4%) [48]. Both species share some recurrent genetic alterations, which creates an opportunity to develop effective treatment strategies.

## 5. Conclusions

Herein, we provide robust data showing that the genomic landscape of AS varies according to the site of origin. The particularities likely represent a different etiologic phenomenon and biologic behavior. For this reason, we need further studies of retrospective cohorts to confirm and expand on the therapeutic implications. Additionally, these findings should be accounted for when designing prospective trials for AS. Finally, incorporating similar genomic testing in the correlative studies of prospective trials can help us build practical predictive tools to combat this deadly aggressive disease.

## Figures and Tables

**Figure 1 cancers-13-04816-f001:**
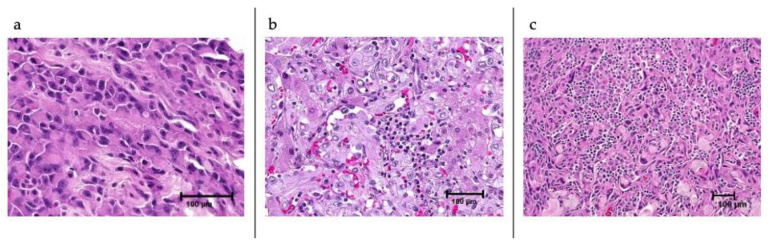
B cell abundance and PDL-1 positivity are present in cases with grade 2 and 3 inflammatory infiltrate by hematoxylin and eosin (H&E). Here, we illustrate the spectrum of the density of inflammation within angiosarcomas. (**a**) Grade 1—<5% of cells are inflammatory cells. (**b**) Grade 2—<30% of cells are inflammatory cells. (**c**) Grade 3—>30% of cells are inflammatory cells.

**Figure 2 cancers-13-04816-f002:**
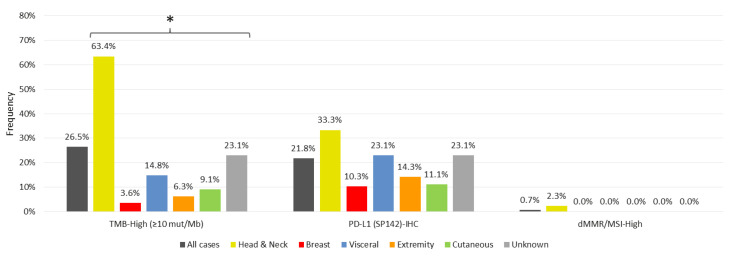
Immunotherapy response biomarkers vary according to the primary site. * Head and neck angiosarcoma cases have a higher predominance of TMB-High (>10 muts/MB), with *p* < 0.0001. In addition, PDL-1 positivity is present at the different sites. AS cases rarely are dMMR/MSI-High.

**Figure 3 cancers-13-04816-f003:**
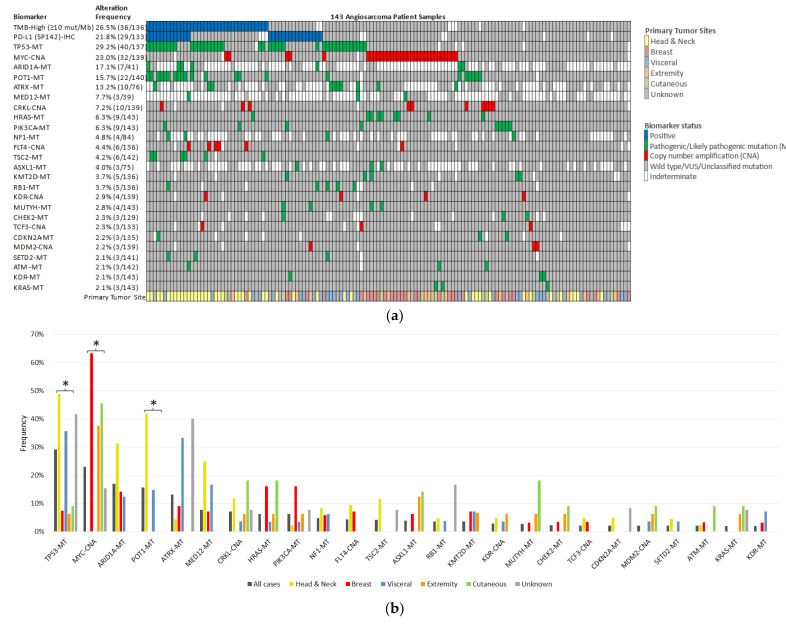
The genomic landscape of angiosarcoma shows a distinct pattern according to the primary site. (**a**) Oncoprint for the entire cohort of 143 cases showing the most common alterations: *TP53* (29%), *MYC* amplification (23%), *ARID1A* (17%), *POT1* (16%), and *ATRX* (13%). (**b**) Genetic alterations vary by primary site. *TP53* and *POT1* are significantly higher in H/N-AS; *MYC* amplification is primarily seen in B-AS. * *p* < 0.0001.

**Figure 4 cancers-13-04816-f004:**
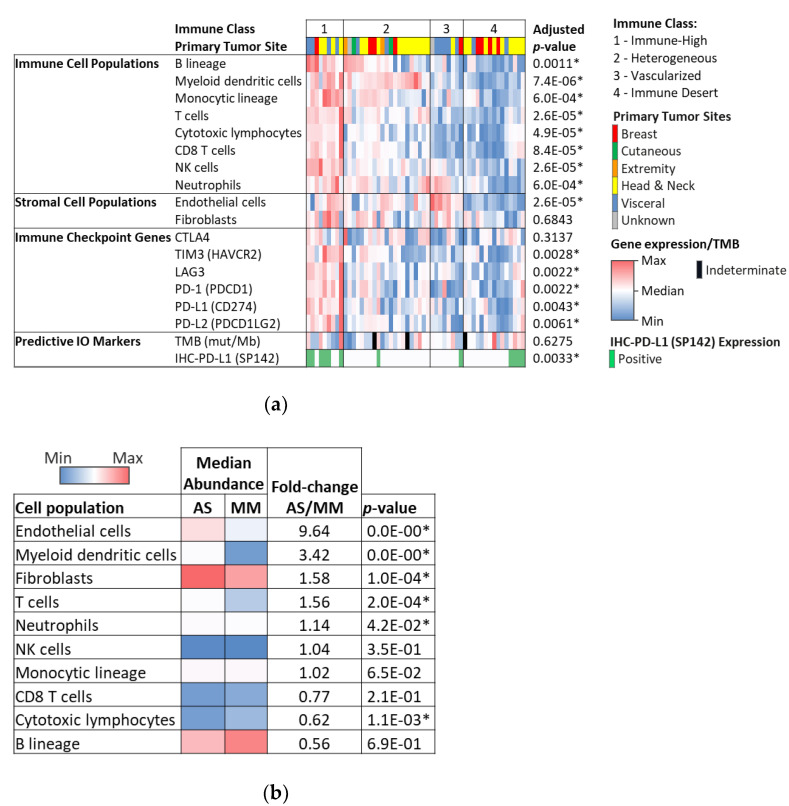
(**a**) Microenvironment phenotype in angiosarcoma. Immune-High phenotype is present in 13% of cases and seen among different primary sites, and this phenotype follows the expression of immune checkpoint genes. (**b**) Comparison of tumor microenvironment between angiosarcoma and melanoma showing a similar abundance of immune cells. * *p* value < 0.05.

**Figure 5 cancers-13-04816-f005:**
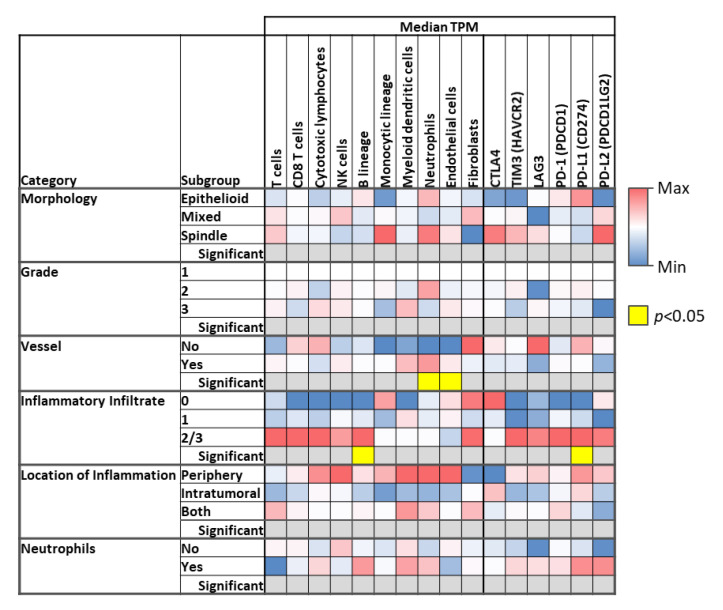
The presence of a grade 2 or 3 inflammatory infiltrate observed on H&E microscopy correlates significantly with higher B cell abundance and PDL-1 expression. In addition, the expression of other immune checkpoint-related genes (CTLA4, TIM3, LAG3, PD1, PD-L1, PD-L2) follows that of the immune cells.

**Table 1 cancers-13-04816-t001:** Primary site distribution and histologic characteristics of cases.

Angiosarcoma Subgroup	All	Head and Neck	Breast	Visceral	Extremity	Cutaneous	Unknown	*p*-Value
Sample size, *N* (%)	143 (100%)	44 (30.8%)	31 (21.7%)	28 (19.6%)	16 (11.2%)	11 (7.7%)	13 (9.1%)	
Morphology								
Epithelioid	46 (32.9%)	19 (43.2%)	8 (26.7%)	9 (32.1%)	3 (18.8%)	3 (30.0%)	4 (36.4%)	0.16
Spindle	9 (6.4%)	0 (0.0%)	3 (10.0%)	4 (14.3%)	0 (0.0%)	1 (10.0%)	1 (9.1%)	
Mixed	85 (60.7%)	25 (56.8%)	19 (63.3%)	15 (53.6%)	13 (81.3%)	6 (60.0%)	6 (54.5%)	
Grade								
1	2 (1.4%)	0 (0.0%)	1 (3.3%)	1 (3.6%)	0 (0.0%)	0 (0.0%)	0 (0.0%)	0.52
2	78 (55.7%)	21 (47.7%)	21 (70.0%)	15 (53.6%)	8 (50.0%)	5 (50.0%)	8 (72.7%)	
3	60 (42.9%)	23 (52.3%)	8 (26.7%)	12 (42.9%)	8 (50.0%)	5 (50.0%)	3 (27.3%)	
Vessel formation								
Yes	117 (83.6%)	35 (79.5%)	28 (93.3%)	23 (82.1%)	12 (75.0%)	9 (90.0%)	9 (81.8%)	0.43
No	23 (16.4%)	9 (20.5%)	2 (6.7%)	5 (17.9%)	4 (25.0%)	1 (10.0%)	2 (18.2%)	
Inflammatory infiltrate								
0	8 (5.7%)	1 (2.3%)	2 (6.7%)	3 (10.7%)	2 (12.5%)	0 (0.0%)	0 (0.0%)	0.11
1	105 (75.0%)	31 (70.5%)	28 (93.3%)	19 (67.9%)	11 (68.8%)	8 (80.0%)	7 (63.6%)	
2	25 (17.9%)	10 (22.7%)	0 (0.0%)	6 (21.4%)	3 (18.8%)	2 (20.0%)	4 (36.4%)	
3	2 (1.4%)	2 (4.5%)	0 (0.0%)	0 (0.0%)	0 (0.0%)	0 (0.0%)	0 (0.0%)	
Location of infiltrate								
Periphery	8 (6.1%)	2 (4.7%)	1 (3.6%)	3 (12.0%)	1 (7.1%)	0 (0.0%)	1 (9.1%)	0.73
Intratumoral	31 (23.5%)	11 (25.6%)	4 (14.3%)	6 (24.0%)	4 (28.6%)	3 (30.0%)	3 (27.3%)	
Both	92 (69.7%)	30 (69.8%)	23 (82.1%)	16 (64.0%)	9 (64.3%)	7 (70.0%)	7 (63.6%)	
Neutrophils present								
Yes	30 (22.7%)	11 (25.6%)	3 (10.7%)	8 (32.0%)	5 (35.7%)	1 (10.0%)	2 (18.2%)	0.29
No	102 (77.3%)	32 (74.4%)	25 (89.3%)	17 (68.0%)	9 (64.3%)	9 (90.0%)	9 (81.8%)	

Note: Four samples (one breast, one cutaneous, and two unknown) did not have hematoxylin and eosin (H&E) slides available for review.

## Data Availability

Data is contained within the article.

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
