# Peer review of "Genomic Landscape of Angiosarcoma: A Targeted and Immunotherapy Biomarker Analysis"

_cancers, 2021, doi:10.3390/cancers13194816_

Round 1
Reviewer 1 Report
This is an interesting manuscript describing the genomic landscape of the different angiosarcoma subtypes. The manuscript is well written, has several strengths including the results of a large subset of patients with NGS data of almost 600 patients, and WES data for around 50 patients. The data showing the differences in immune profiles between the different AS subtypes is also strengthening the manuscript.
However, I do have two concerns:
- In the manuscript, there is no information regarding which angiosarcoma were in fact radiation associated angiosarcoma. This is quite essential, since both the molecular profiles as responses to therapy/biological behaviour is quite distinct in radiation associated angiosarcoma compared to others. Are most breast angiosarcomas in this cohort radiation associated (the Myc amplifications suggest this)? How many of the cuteanous/extremy AS were? I would make a seperate group for the radiation associated AS - despite there location. I would therefor suggest to cahge the categories into: UV induced/HN AS - radiation associated AS - Stewert Treves AS - primary/idiopatic AS.
- Adding clinical data and correlate the NGS and especially the WES results with oncological outcome would significantly improve the impact of the data.
Author Response
Dear Reviewer,
Thank you for the time assessing our manuscript. We appreciate the input you have given to our work which will substantially enhance the quality of the manuscript. Below you will find an itemized response to each critique.
Our cohort comprises NGS data of 143 patients and WTS was available for 53 of them. We agree with you that the data of association to radiation therapy is fundamental. Unfortunately, this annotation is unavailable to us as limited clinical characteristics were obtained from the requisition forms used to order the NGS. However, robust data from previous cohorts shows that MYC amplification is almost exclusively seen in RT-associated breast AS cases. Most cases of breast AS having MYC amp is likely a reflection of that, as RT-associated AS of breast is more common than primary breast AS.
Thank you for pointing out the presence of MYC amplification at other anatomic sites, it is important and clinically meaningful to describe. Overall, MYC amplification is present in 23% of the cohort and 63% of breast AS; we have added the following to the results section line 211: “Interestingly, MYC amplification is also seen in 45.5% (5/11) of cutaneous cases at other locations than H/N or breast, and 37.5% (6/16) of cases of extremity AS, but not seen in H/N or visceral AS cases. MYC amp has been described in radiation-associated AS of the breast [17-19]; future studies are needed to investigate further whether MYC amplification is an etiologic factor at other sites of origin. Unfortunately, our cohort does not annotate whether the patient had a prior history of radiation exposure.”
We have a similar limitation for the clinical data of outcomes. We support your point of view that the impact of analyzing genomic data of rare tumors would be more meaningful if we have correlations with clinical behavior and responses to different treatment strategies. In our discussion section (Line 367 - 370 and 374 - 378), we underscored this point.
“Over 60% of cases of B-AS show MYC amplification. In our cohort, we do not have data about the association of the cases with prior radiation therapy. However, there is strong evidence from prior reports that MYC amplification is almost exclusively seen in radiation-associated AS cases [11,12].”… “Finally, we acknowledge that an important constraint of our study is the limited clinical data obtained from the requisition forms submitted by the ordering physician. In studying rare tumors like AS, this type of genomic analysis should be performed in retrospective and prospective cohorts with available clinical data of responses to the different therapeutic strategies to understand the therapeutic implications further.”
Future studies are planned to abstract clinical information on the angiosarcoma patients from each of these sites in order to better identify clinically relevant biomarkers from this data set.
Reviewer 2 Report
Manuscript needs major revision
Abstract should be clear and concise with clear conclusion what mutations are most important for the angiosarcoma. Authors should state in the Abstract that it is retrospective study
Authors should clearly state in the Abstract that this study was performed on sample from the head and neck, breast sarcoma, visceral, extremity, cutaneous and how they apply findings on other cancers
Authors should state is this retrospective or prospective study
Authors need to present what therapeutic strategies they see as a solution in angio sarcoma treatment
In the introduction elaborate more on angiosarcoma clinics and genomics
Elaborate more on microenvironment and its genomics and signaling elements
Fig1 In Fig Legend explain why inflammatory cell are important for the sarcoma, genomics and biomarkers
Fig 2. In Figure legend elaborate more on dMMR/MSI, PD-L1, and
Tumor Mutation Burden in the immunotherapy
Fig 3. In Fig Legend elaborate more on most important genes expressed
Fig 4. explain why for comparison was taken sarcoma and melanoma? Present heat map for sarcoma cells with explanation of most expressed genes important in clinics
Fig 5 needs detailed explanation on gene expression
In the Discussion explain importance of sarcoma microenvironment genomics for the clinics. Explain in the details role of mutations found in sarcoma for the therapy and diagnostics.
Author Response
Dear Reviewer,
Thank you for the time assessing our manuscript. We appreciate the input you have given to our work which will substantially enhance the quality of the manuscript. Below you will find an itemized response to each critique.
Manuscript needs major revision
Abstract should be clear and concise with clear conclusion what mutations are most important for the angiosarcoma. Authors should state in the Abstract that it is retrospective study
Authors should clearly state in the Abstract that this study was performed on sample from the head and neck, breast sarcoma, visceral, extremity, cutaneous and how they apply findings on other cancers
Response
Revised Abstract
Thank you for the constructive critique. As suggested we have clarified the abstract to make it more concise and to point out specific important mutations.
Abstract: We performed a comprehensive retrospective analysis of angiosarcoma (AS) genomic biomarkers and their associations with the site of origin. We studied a cohort of 143 cases of AS profiled by Caris Life Sciences. The number of cases by location were head and neck (n=44, 31%), breast (n=31, 22%), extremity (n=16, 11%), viscera (n=28, 20%), skin at other locations (n=11, 8%), and unknown (n=13, 9%). Data of Next Generation Sequencing (NGS) with a 592 gene panel was available for the entire cohort. Fifty-three cases had Whole Exome Sequencing (WES) data, which we used to study the microenvironment phenotype. We included Immunotherapy (IO) response biomarkers: Tumor Mutation Burden (TMB), Microsatellite Instability (MSI), and PD-L1 status. IO-response markers were present in 36.4% of the cohort and 65% of head and neck AS (H/N-AS) (p<0.0001). H/N-AS cases had predominantly mutations in TP53 (50.0%, p=0.0004), POT1 (40.5%, p<0.0001) and ARID1A (33.3%, p=0.5875). In breast AS, leading alterations were MYC amplification (63.3%, p<0.0001), HRAS (16.1%, p=0.0377), and PI3KCA (16.1%, p=0.2352). At other sites, conclusions are difficult to generate due to the small number of cases. A microenvironment with a high immune signature, associated with a better response to IO, was present in 13% of the cases. This signature is evenly distributed among different primary sites. We found that the molecular biology for AS varies significantly according to the primary site. Our findings can facilitate the design and optimization of therapeutic strategies for AS to overcome resistance to IO and targeted therapies.
Authors should state is this retrospective or prospective study
This oversight was corrected in the first sentence of the abstract.
Authors need to present what therapeutic strategies they see as a solution in angiosarcoma treatment
Response
Thank you for the input, we have elaborated more about this in the discussion section.
Please see discussion lines 338 to 358 on therapeutic implications of the MCP counter method and lines 359 to 380 about implications of genetic mutations.
“However, not a single phenotype across solid tumors has yielded similar prognostic and predictive capabilities. Whereas in AS, the phenotype with an abundance of B cells results in similar predictive capabilities of what has been seen for STS warrants further study. Additionally, the dynamics of the microenvironment upon treatment could potentially have better predictive capabilities. In melanoma, a highly immunogenic tumor where IO is active, INF-gamma driven infiltration of CD8+ lymphocytes upon treatment predicts responses[26,29]. In our cohort, we described a similar microenvironment to that of melanoma. Future studies should further examine the predictive capabilities of microenvironment analysis in AS to move forward to incorporating this method into clinical practice. WTS is commonly performed when profiling solid tumors; however, immune cell abundance and microenvironment cell counter results are not routinely reported to treating physicians. As more evidence of the predictive capabilities of this method emerges, we may be able to incorporate this information into clinical decision-making as we move forward in tailoring effective therapies for rare tumors. Importantly, we reviewed the H&E slides of 138 of the patients (4 cases had no available H&E slide) and described the tumor inflammatory infiltrate. We saw that B-cell abundance by WTS and PDL-1 expression was associated with the presence of an inflammatory infiltrate of grade 2 or 3 as assessed by light microscopy. Thus, we could optimize this strategy by incorporating immunohistochemistry to compare the predictive phenotypes determined by the MCP-counter method.
Certain genetic alterations of AS are more common at specific primary sites. Consequently, further studies to overcome IO resistance and increase the effectiveness of targeted therapies accounting for specific alterations are warranted. Therapeutic strategies that target TP53, POT1, and ARID1A could be of value in H-N AS. In cases with a TP53 mutation, we found that 67% had concomitant markers of IO response. Therefore, it may be important to investigate strategies using agents that target TP53 function, especially for cases of H-N AS that are resistant to IO. ARID1A is one of the most common alterations encountered in our cohort. Being part of the chromatin remodeling complex SWI/SNF, its deficiency results in EZH2 overactivity. In epithelioid sarcoma, we have observed success with the use of EZH2 inhibitor tazemetostat. Therefore, prospective evaluation of tazemetostat in ARID1A mutated AS should be considered[30]. Over 60% of cases of B-AS show MYC amplification. In our cohort, we do not have data about the association of the cases with prior radiation therapy. However, there is strong evidence that MYC amplification is almost exclusively seen in radiation-associated AS cases [11,12]. Other common mutations found in B-AS are PI3K and HRAS; other reports indicate that these are primarily found in cases of primary B-AS. These differences should be accounted for when devising differential strategies to treat RT-associated B-AS and primary B-AS. “
In the introduction elaborate more on angiosarcoma clinics and genomics
We appreciate this suggestion from the reviewer, we agree that the distinct clinical behavior at different sites of origin is an important background anteceding the different genomic findings. Therefore, we incorporated the following paragraph in the introduction:
“At the different sites of origin, cases of AS show different clinical features and prognosis. Most common AS location is head and neck AS (H/N-AS), followed by breast AS (B-AS), visceral, other cutaneous sites, and the extremities[12]. The majority of cases of AS occur sporadically (primary) or are related to radiation therapy or chronic lymphedema (secondary) [8,9]. A French retrospective multicenter study of 204 patients reported that visceral (heart, liver, and spleen) and primary bone sites were associated with worse prognosis[14]. Also showing biological differences, a study of 200 AS cases from China described that the worst prognosis is seen in H/N-AS (5-year OS of 28%), followed by visceral (37%) and B-AS (87%)[12]. Evidence shows that patients with secondary breast AS (B-AS) have a more aggressive tumor phenotype and worse survival outcome than patients with primary B-AS[10,11]. A study of over 470 patients extracted from the SEER database described that secondary B-AS appears in older patients, presents with more locally advanced stage (57% vs. 18%) and high grade (58% vs. 32%). In this cohort, the median OS was 93 months for primary B-AS and 32 months for secondary B-AS[15].”
Most important data on AS genomics is described in lines 78 to 91.
Along with the differences in clinical behavior, some small cohorts in the literature show genomic differences within AS. The first identified genetic alteration was KDR (AKA VEGFR2) which harbors point mutations in 10% of primary or secondary B-AS [16]. Other recurrent reported alterations are TP53, PI3KCA, POT1, RAS, BRAF, PTPRB, PLCG1 and APC [2,11,17]. Some mutations appear to be distinct to cases of primary and secondary AS. For MYC amplification, reported in 50 to 100% of radiation-associated AS cases but not in primary AS [18]–[20]. Most recently, Whole Exome Sequencing (WES) results of 47 samples from 36 patients self-registered to the Angiosarcoma Project were published. In this cohort, the authors reported that TP53 and KDR mutations are mutually exclusive, with 89% of KDR mutations in primary B-AS compared to 82% of TP53 present in non-primary B-AS (p=0.02). Nine out of ten PI3KCA alterations were also seen in primary B-AS (P=0.0003) [11]. Despite sequencing techniques allowing the identification of recurrent somatic genetic abnormalities, the rarity of AS challenges our efforts to establish strong associations with the site of origin, etiology, and therapeutic implications.
Elaborate more on microenvironment and its genomics and signaling elements
As per reviewer request, additional background on microenvironment in STS is described in the Introduction line 108 to 118 and in methods line 132 to 139:
“Transcriptomic analysis is now available to estimate the relative abundance of immune and stromal cells within tumor samples. Using this technique, Petiprez et al. described a classification of soft tissue sarcomas based on their tumor microenvironment. In the SARC028 trial for the use of PDL-1 blockage for STS, they identified that an immune-rich microenvironment, particularly a B cell abundance, correlated with better response rate and improved PFS[22]. Interestingly, the overall TMB appeared similar across all classes of microenvironment phenotypes. In other histologies, microenvironment analysis also shows predictive capabilities for IO and other targeted therapies. For example, in renal cell carcinoma, gene expression signatures of angiogenesis, T-effector, and myeloid cells are predictive of PFS for IO alone or combined with anti-VEGF[23]. Whereas these methods can be applied similarly to patients with AS needs further investigation”.
“WTS was performed on 53 tumors and used for microenvironment cell population (MCP)-counter analysis as described by Becht et al. [24]. First, using transcriptomic markers (TM), estimating a cell population of interest from the sample is possible. TM are gene expression features expressed in one and only one cell population. The method generates an abundance score for CD3+ T cells, CD8+ T cells, cytotoxic lymphocytes, NK cells, B lymphocytes, monocyte lineage cells, endothelial cells, and fibroblasts[24]. Next, we identified subgroups based on tumor microenvironment profiles by hierarchical clustering of MCP-counter Z-scores [22]”.
Fig1 In Fig Legend explain why inflammatory cell are important for the sarcoma, genomics and biomarkers
We have added additional information on the importance of inflammatory cells for sarcoma.
Figure 1. B cell abundance and PDL-1 positivity is mostly seen in cases with grade 2 and 3 inflammatory infiltrate by H&E. Here, we illustrate the spectrum of the density of inflammation within angiosarcomas. a) grade 1 - <5% of cells are inflammatory cells. b) grade 2 - <30% of cells are inflammatory cells. c) grade 3 - >30% of cells are inflammatory cells.
Fig 2. In Figure legend elaborate more on dMMR/MSI, PD-L1, and Tumor Mutation Burden in the immunotherapy.
Figure 2. Immunotherapy response biomarkers vary according to the primary site. ** Head and neck angiosarcoma cases have a significantly higher predominance of TMB-High (>10 muts/MB) with p<0.0001. In addition, PDL-1 positivity is present among the different sites. AS cases rarely are dMMR/MSI- High.
Here, in the figure, we will not make any comments about these markers and immunotherapy response. Although the presence of these markers suggests IO activity, we cannot draw this conclusion from our cohort. Moreover, as in other tumors, classic biomarkers of IO response have predictive limitations.
Fig 3. In Fig Legend elaborate more on most important genes expressed
Figure 3. Figure 3. The genomic landscape of angiosarcoma shows a distinct pattern according to the primary site. a) Oncoprint for the entire cohort of 143 cases showing the most common alterations: TP53 (29%), MYC amp (23%), ARID1A (17%), POT1 (16%), and ATRX (13%). b) Genetic alterations vary by primary site. TP53 and POT1 are significantly higher in H-N AS; MYC amp is primarily seen in B-AS**p<0.0001
Fig 4. explain why for comparison was taken sarcoma and melanoma? Present heat map for sarcoma cells with explanation of most expressed genes important in clinics
Line 258 - 262: “Next, we compared the microenvironment of the AS cohort with a cohort of melanoma (n=1255). Of all tumors in humans, melanomas have one of the highest-burden of somatic genetic alterations[25]. Moreover, particularly in cutaneous melanoma, an extensive CD8+ T-cell infiltration has been described and associated with better prognosis and response to IO [26]".
The most common altered genes are shown in figure 3.
Fig 5 needs detailed explanation on gene expression
Fig. 5 The presence of a grade 2 or 3 inflammatory infiltrate observed on H&E microscopy correlates significantly with higher B cell abundance and PDL-1 expression. The expression of other immune checkpoint-related genes (CTLA4, TIM3, LAG3, PD1, PD-L1, PD-L2) follows that of the immune cells.
In the Discussion explain importance of sarcoma microenvironment genomics for the clinics. Explain in the details role of mutations found in sarcoma for the therapy and diagnostics.
Please see discussion line 331 to 358 for the implications of microenvironment in the clinics. We have mentioned above the therapeutic implications of the mutations. However, we decided not to expand in specific agents because strategies for targeting those genes are extensive and deserve a comprehensive analysis which is not the purpose of our manuscript.
“Here, we also describe a microenvironment with a high immune signature and abundance of B-cells is present in about 13% of the cases and evenly distributed among different primary sites. A signature of B-cell lineage abundance regardless of high or low CD8+ T cell infiltration appears to predict response to PD1 blockade and PFS in soft-tissue-sarcomas[22]. Interestingly, in this immune-high group, the cases had overall a low TMB. The contribution of each of these factors and the dynamic microenvironment changes to IO response are still unknown. In other solid tumors, distinct microenvironment characteristics show predictor capabilities to IO and other targeted therapies[19,20]. However, not a single phenotype across solid tumors has yielded similar prognostic and predictive capabilities. Whereas in AS, the phenotype with an abundance of B cells results in similar predictive capabilities of what has been seen for STS warrants further study. Additionally, the dynamics of the microenvironment upon treatment could potentially have better predictive capabilities. In melanoma, a highly immunogenic tumor where IO is active, INF-gamma driven infiltration of CD8+ lymphocytes upon treatment predicts responses[26,29]. In our cohort, we described a similar microenvironment to that of melanoma. We need to examine further the predictive capabilities of microenvironment analysis in AS to move forward to incorporating this method into clinical practice. WTS is commonly performed when profiling solid tumors; however, immune cell abundance and microenvironment cell counter results are not routinely reported to treating physicians. As more evidence of the predictive capabilities of this method comes out, we need to incorporate it into clinical decision-making as we move forward in tailoring effective therapies for rare tumors. Importantly, we reviewed the H&E slides of 138 of the patients (4 cases had no available H&E slide) and described the tumor inflammatory infiltrate. We saw that B-cell abundance by WTS and PDL-1 expression was associated with the presence of an inflammatory infiltrate of grade 2 or 3 as assessed by light microscopy. Thus, we could optimize this strategy by incorporating immunohistochemistry to compare the predictive phenotypes determined by the MCP-counter method. “
Reviewer 3 Report
I would like to congratulate the authors on this relevant, very interesting and well-written manuscript.
A remarkably high number of patients with a very rare disease (with angiosarcoma) have been extensively studied for genetic changes and potential therapeutic consequences. In addition, a reference pathological re-assessment of the histological samples was carried out. The description of the methodology is understandable and adequate. The results of the investigations carried out are very interesting and relevant. The classification of the results in context and their discussion as well as the drawn conclusions are well-founded and easy to understand. All in all a really excellent paper!
Author Response
Dear Reviewer,
Thank you for the time assessing our manuscript, we are excited to contribute with these findings and move forward in designing more effective therapies for AS.
Reviewer 4 Report
The paper of Espejo-Freire and colleagues is another article on the genomic landscape in angiosarcoma (AS) that has been written over the past few years. The efficacy of current treatments of AS, including immunotherapy and targeted therapy, is limited. A better understanding of the genetic and epigenetic determinants of treatment’s efficacy could improve outcomes and identify new therapeutic targets. The value of this study is the large group of 143 patients analyzed with a rare cancer such as AS. The authors have demonstrated at the genomic level, that AS are heterogenous, and that their molecular biology varies significantly dependently to the primary site what may reflect their different origins and biology.
I have no major objections to this article, however, some analysis could be added. The authors have not performed an analysis of a dominant UV light exposure mutational signature which is found in melanoma and other skin tumors and may, therefore, be another similarity with tumor types known to respond to PD-1/PD-L1 blockade. The high prevalence of UV-induced genomic variants appears to be a specific hallmark of face and neck angiosarcoma, as recently described by the patient-partnered Angiosarcoma Project initiators (Corrie A. Painter at al. 2020) as well as Amelie Boichard et al. (Genome Medicine 2020). Therefore, if it was confirmed that the subset of AS patients of this localization might benefit from treatment with checkpoint inhibition.
Author Response
Dear Reviewer,
Thank you for the time assessing our manuscript. We appreciate the input you have given to our work which will substantially enhance the quality of the manuscript. Below you will find an itemized response to each critique.
We have made every effort to incorporate the mutational signatures by applying the COSMIC signature analysis to our samples. Unfortunately, mutational signature analysis has predominately used exome sequencing. Even more, whole-genome sequencing probably adds to the power for signature decomposition. In our cohort, the sequence coverage by the 592 gene panel is limited and thus yields unreliable results. We added this limitation to the discussion section line 313 to 322.
“In these prior reports, UV light mutational signatures have been described through whole genome and whole exome sequencing analysis. Unfortunately, in our cohort, the data comes from a specific 592 gene panel that involves sequencing a significantly smaller part of the genome. As such, we have limited power to present the results of mutational signatures. However, the consistency of other findings suggests that a UV mutational signature is likely to be found in cases of H-N AS associated with high TMB. These factors may explain why H-N/AS cases benefit more from IO. The retrospective series by Florou et al. showed that most responders were cases of H-N AS (4/5 cases)[10]. Similarly, the 2 cases of outstanding responses noted in the Angiosarcoma Project had TMB High H-N AS[11].”
As stated in Alexandrov et al., 2020, “Mutational signature analysis has predominantly used cancer exome sequences. However, the many-fold-greater numbers of somatic mutations in whole genomes provide substantially increased power for signature decomposition, enabling the better separation of partially correlated signatures and the extraction of signatures that contribute relatively small numbers of mutations. Furthermore, technical artefacts and differences in sequencing technologies and mutation-calling algorithms can themselves generate mutational signatures.”
For this reason, we are not able to incorporate this analysis in our paper. We agree that having this information to compare and confirm the results of other smaller cohorts would have been helpful. However, other results extracted from our study are in line with what has been previously described.
Round 2
Reviewer 2 Report
Manuscript is not corrected as recommended
Authors need Abstract conclusion to be based on finding
And these findings should be discussed leter in the section Discussion
Figure 2 and 3 are representative figures for the manuscript
For example
Abstract conclusion should say: Head and neck has predominant TMB biomarker but rest of loci PDL1 and also
Genomic most common alteration in angiosarcoma are
TP53 (29%), MYC amp (23%), ARID1A (17%), POT1 (16%), and ATRX (13%).
Author Response
Dear Editor,
Thank you for the time assessing our manuscript. We appreciate the input you have given to our work which will substantially enhance the quality of the manuscript. We have added the recommended data by reviewer 2. However, the we are not sure we fully understand, so we have edited the manuscript to the best of our ability. Please do not hesitate to contact us for additional edits or clarification. Below, you will find our response to each critique and comment on round 2, followed at the end by the response to round 1 comments.
Thank you again for the insightful review which has substantially improved the quality of the manuscript.
Sincerely,
The Authors
Reviewer 2 Round 2 Comments
Manuscript is not corrected as recommended
Authors need Abstract conclusion to be based on finding
And these findings should be discussed leter in the section Discussion
Figure 2 and 3 are representative figures for the manuscript
For example
Abstract conclusion should say: Head and neck has predominant TMB biomarker but rest of loci PDL1 and also
Genomic most common alteration in angiosarcoma are
TP53 (29%), MYC amp (23%), ARID1A (17%), POT1 (16%), and ATRX (13%).
Critique
Authors need Abstract conclusion to be based on finding
Abstract conclusion should say: Head and neck has predominant TMB biomarker but rest of loci PDL1 and also
Genomic most common alteration in angiosarcoma are
TP53 (29%), MYC amp (23%), ARID1A (17%), POT1 (16%), and ATRX (13%).
Response
Thank you for your input about including all the most common alterations in the abstract, we have done so and expanded on the therapeutic implications of the findings.
Please see the revised abstract and the new section of the discussion below:
Abstract
We performed a retrospective analysis of angiosarcoma genomic biomarkers and their associations with the site of origin in a cohort of 143 cases. Primary sites were head and neck (31%), breast (22%), extremity (11%), viscera (20%), skin at other locations (8%), and unknown (9%). All cases had Next Generation Sequencing (NGS) data with a 592 gene panel, and 53 cases had Whole Exome Sequencing (WES) data, which we used to study the microenvironment phenotype. Immunotherapy (IO) response biomarkers: Tumor Mutation Burden (TMB), Microsatellite Instability (MSI), and PD-L1 status were the most frequently encountered alteration, present in 36.4% of the cohort and 65% of head and neck AS (H/N-AS) (p<0.0001). In H/N-AS, TMB-High is seen in 63.4% of cases (p<0.0001) and PDL-1 positivity in 33% of cases. The most common genetic alterations were TP53 (29%), MYC amp (23%), ARID1A (17%), POT1 (16%), and ATRX (13%). H/N-AS cases had predominantly mutations in TP53 (50.0%, p=0.0004), POT1 (40.5%, p<0.0001) and ARID1A (33.3%, p=0.5875). In breast AS, leading alterations were MYC amplification (63.3%, p<0.0001), HRAS (16.1%, p=0.0377), and PI3KCA (16.1%, p=0.2352). At other sites, conclusions are difficult to generate due to the small number of cases. A microenvironment with a high immune signature, previously associated with IO response, was evenly distributed in 13% of the cases at different primary sites. Our findings can facilitate the design and optimization of therapeutic strategies for AS.
Critique
And these findings should be discussed leter in the section Discussion
Response
Discussion
“Certain genetic alterations of AS are more common at specific primary sites. Consequently, further studies to overcome IO resistance and increase the effectiveness of targeted therapies accounting for specific alterations are warranted. Therapeutic strategies that target TP53, POT1, and ARID1A could be of value in H-N AS. TP53 tumor suppressor activity triggers cell cycle arrest, dead by apoptosis, and senescence by regulation of multiple pathways. Even though TP53 is widely mutated in cancer, targeting it has been challenging. Recently, the study of small molecules to reestablish the activity of mutant p53 has shown promising results. Particularly, APR-246 (eprenetapopt), which refolds mutant p53 to induce p53 target genes, demonstrated clinical activity in myeloid malignancies[30,31]. This promising strategy is currently under investigation in combination with IO, chemotherapy, and other targeted agents. In cases with a TP53 mutation, we found that 67% had concomitant markers of IO response. Therefore, it may be essential to investigate if the use of this molecule could overcome IO resistance in H-N. ARID1A is one of the most common alterations encountered in our cohort and is also seen predominately in H/N-AS. Being part of the chromatin remodeling complex SWI/SNF, its deficiency results in EZH2 overactivity. In epithelioid sarcoma, we have observed success using EZH2 inhibitor tazemetostat. This strategy has shown efficacy in ARID1A mutated ovarian and endometrioid cell carcinomas[32]. Therefore, prospective evaluation of tazemetostat in ARID1A mutated AS should be considered. Lastly, POT1 is involved in telomere maintenance, and its regular activity results in cell aging and apoptosis. Three percent of malignancies have POT 1 mutation; however, its prevalence is higher in AS (23%) and is described among the top predisposition genes for familial melanoma and cardiac AS[34]. In our cohort, it was predominately present in cases of H/N-AS. Unfortunately, the therapeutic role of POT1 inhibition in cancer is currently unknown.
The molecular alterations in B-AS give additional opportunities for other therapeutic strategies. Over 60% of cases of B-AS show MYC amplification. In our cohort, we do not have data about the association of the cases with prior radiation therapy. However, strong evidence shows that MYC amplification is almost exclusively seen in radiation-associated AS[18, 20, 34]. MYC proteins coordinate transcription, DNA replication, and cell cycle progression. Strategies to target cell cycle by CDK inhibition are showing promising results in MYC amplified tumors. In neuroblastoma cell lines, CDK/CDK1 inhibitors show an ability to downregulated MYC [36]. Recently, fadraciclib, a potent inhibitor of CDK9 showed an ability to repress MYC and is currently in early-phase clinical trials for solid tumors and hematologic malignancies. [37], [38]. Additionally, bromodomain inhibitors (BET) have proven efficacy in MYC amplified lymphoma. Its use in combination with CDK2 inhibitor is being studied in MYC driven medulloblastoma[39], [40][41]. Finally, in ovarian cancer, MYC amplification predicted synergistic benefit from a combination of PARP inhibition with olaparib and CDK-4 inhibition with Palbociclib[42]. These strategies deserve further studies in MYC amplified AS. Other common mutations found in B-AS are PI3K and HRAS. Prior reports indicate that PI3KCA mutations are primarily found in cases of primary B-AS. PI3KCA inhibition is efficacious PI3KCA mutated breast cancer. These mutations found in primary breast AS suggest that site of origin may predispose to overactive PI3KCA driven AS. Here evaluation of the use of alpelisib or other PI3KCA inhibitor is necessary for cases of primary breast AS[43]. These differences should be accounted for when devising differential strategies to treat RT-associated B-AS and primary B-AS. On the other hand, mutations in RAS usually result in downstream activation of MAPK and PI3K. Several compounds targeting downstream effects of RAS, such as MEK, AKT, and PI3K are available and should be considered in these cases. This agent's synergistic effects in combination with IO are also being studied [44], [45].
Finally, we acknowledge that an important constraint of our study is the limited clinical data obtained from the requisition forms submitted by the ordering physician. Because preclinical models of AS are limited, this type of genomic analysis should be performed in retrospective and prospective cohorts with available clinical data of responses to the different therapeutic strategies to understand the therapeutic implications further. Importantly, a recent study showed similarities of AS genomic landscape to that of canine hemangiosarcoma. In canine hemangiosarcoma recurrent oncogenic mutations were TP53 (66%), PI3CA (46%), NRAS (24%), PTEN(6%), and PLCG1(4%)[46]. Both species share some recurrent genetic alterations, which opens a big opportunity to develop effective treatment strategies.
Comment
Figure 2 and 3 are representative figures for the manuscript
Response
We agree these are reprentative figures and have included more detailed information in the abstract and discussion.
Reviewer 2 Round 1
Manuscript needs major revision
Abstract should be clear and concise with clear conclusion what mutations are most important for the angiosarcoma. Authors should state in the Abstract that it is retrospective study
Authors should clearly state in the Abstract that this study was performed on sample from the head and neck, breast sarcoma, visceral, extremity, cutaneous and how they apply findings on other cancers
Response
Revised Abstract
Thank you for the constructive critique. As suggested, we have clarified the abstract to make it more concise and to point out specific important mutations.
Abstract: We performed a comprehensive retrospective analysis of angiosarcoma (AS) genomic biomarkers and their associations with the site of origin. We studied a cohort of 143 cases of AS profiled by Caris Life Sciences. The number of cases by location were head and neck (n=44, 31%), breast (n=31, 22%), extremity (n=16, 11%), viscera (n=28, 20%), skin at other locations (n=11, 8%), and unknown (n=13, 9%). Data of Next Generation Sequencing (NGS) with a 592 gene panel was available for the entire cohort. Fifty-three cases had Whole Exome Sequencing (WES) data, which we used to study the microenvironment phenotype. We included Immunotherapy (IO) response biomarkers: Tumor Mutation Burden (TMB), Microsatellite Instability (MSI), and PD-L1 status. IO-response markers were present in 36.4% of the cohort and 65% of head and neck AS (H/N-AS) (p<0.0001). H/N-AS cases had predominantly mutations in TP53 (50.0%, p=0.0004), POT1 (40.5%, p<0.0001) and ARID1A (33.3%, p=0.5875). In breast AS, leading alterations were MYC amplification (63.3%, p<0.0001), HRAS (16.1%, p=0.0377), and PI3KCA (16.1%, p=0.2352). At other sites, conclusions are difficult to generate due to the small number of cases. A microenvironment with a high immune signature, associated with a better response to IO, was present in 13% of the cases. This signature is evenly distributed among different primary sites. We found that the molecular biology for AS varies significantly according to the primary site. Our findings can facilitate the design and optimization of therapeutic strategies for AS to overcome resistance to IO and targeted therapies.
Authors should state is this retrospective or prospective study
This oversight was corrected in the first sentence of the abstract.
Authors need to present what therapeutic strategies they see as a solution in angiosarcoma treatment
Response
Thank you for the input, we have elaborated more about this in the discussion section.
Please see discussion lines 338 to 358 on therapeutic implications of the MCP counter method and lines 359 to 380 about implications of genetic mutations.
“However, not a single phenotype across solid tumors has yielded similar prognostic and predictive capabilities. Whereas in AS, the phenotype with an abundance of B cells results in similar predictive capabilities of what has been seen for STS warrants further study. Additionally, the dynamics of the microenvironment upon treatment could potentially have better predictive capabilities. In melanoma, a highly immunogenic tumor where IO is active, INF-gamma driven infiltration of CD8+ lymphocytes upon treatment predicts responses[26,29]. In our cohort, we described a similar microenvironment to that of melanoma. Future studies should further examine the predictive capabilities of microenvironment analysis in AS to move forward to incorporating this method into clinical practice. WTS is commonly performed when profiling solid tumors; however, immune cell abundance and microenvironment cell counter results are not routinely reported to treating physicians. As more evidence of the predictive capabilities of this method emerges, we may be able to incorporate this information into clinical decision-making as we move forward in tailoring effective therapies for rare tumors. Importantly, we reviewed the H&E slides of 138 of the patients (4 cases had no available H&E slide) and described the tumor inflammatory infiltrate. We saw that B-cell abundance by WTS and PDL-1 expression was associated with the presence of an inflammatory infiltrate of grade 2 or 3 as assessed by light microscopy. Thus, we could optimize this strategy by incorporating immunohistochemistry to compare the predictive phenotypes determined by the MCP-counter method.
Certain genetic alterations of AS are more common at specific primary sites. Consequently, further studies to overcome IO resistance and increase the effectiveness of targeted therapies accounting for specific alterations are warranted. Therapeutic strategies that target TP53, POT1, and ARID1A could be of value in H-N AS. In cases with a TP53 mutation, we found that 67% had concomitant markers of IO response. Therefore, it may be important to investigate strategies using agents that target TP53 function, especially for cases of H-N AS that are resistant to IO. ARID1A is one of the most common alterations encountered in our cohort. Being part of the chromatin remodeling complex SWI/SNF, its deficiency results in EZH2 overactivity. In epithelioid sarcoma, we have observed success with the use of EZH2 inhibitor tazemetostat. Therefore, prospective evaluation of tazemetostat in ARID1A mutated AS should be considered[30]. Over 60% of cases of B-AS show MYC amplification. In our cohort, we do not have data about the association of the cases with prior radiation therapy. However, there is strong evidence that MYC amplification is almost exclusively seen in radiation-associated AS cases [11,12]. Other common mutations found in B-AS are PI3K and HRAS; other reports indicate that these are primarily found in cases of primary B-AS. These differences should be accounted for when devising differential strategies to treat RT-associated B-AS and primary B-AS. “
In the introduction elaborate more on angiosarcoma clinics and genomics
We appreciate this suggestion from the reviewer, we agree that the distinct clinical behavior at different sites of origin is an important background anteceding the different genomic findings. Therefore, we incorporated the following paragraph in the introduction:
“At the different sites of origin, cases of AS show different clinical features and prognosis. Most common AS location is head and neck AS (H/N-AS), followed by breast AS (B-AS), visceral, other cutaneous sites, and the extremities[12]. The majority of cases of AS occur sporadically (primary) or are related to radiation therapy or chronic lymphedema (secondary) [8,9]. A French retrospective multicenter study of 204 patients reported that visceral (heart, liver, and spleen) and primary bone sites were associated with worse prognosis[14]. Also showing biological differences, a study of 200 AS cases from China described that the worst prognosis is seen in H/N-AS (5-year OS of 28%), followed by visceral (37%) and B-AS (87%)[12]. Evidence shows that patients with secondary breast AS (B-AS) have a more aggressive tumor phenotype and worse survival outcome than patients with primary B-AS[10,11]. A study of over 470 patients extracted from the SEER database described that secondary B-AS appears in older patients, presents with more locally advanced stage (57% vs. 18%) and high grade (58% vs. 32%). In this cohort, the median OS was 93 months for primary B-AS and 32 months for secondary B-AS[15].”
Most important data on AS genomics is described in lines 78 to 91.
Along with the differences in clinical behavior, some small cohorts in the literature show genomic differences within AS. The first identified genetic alteration was KDR (AKA VEGFR2) which harbors point mutations in 10% of primary or secondary B-AS [16]. Other recurrent reported alterations are TP53, PI3KCA, POT1, RAS, BRAF, PTPRB, PLCG1 and APC [2,11,17]. Some mutations appear to be distinct to cases of primary and secondary AS. For MYC amplification, reported in 50 to 100% of radiation-associated AS cases but not in primary AS [18]–[20]. Most recently, Whole Exome Sequencing (WES) results of 47 samples from 36 patients self-registered to the Angiosarcoma Project were published. In this cohort, the authors reported that TP53 and KDR mutations are mutually exclusive, with 89% of KDR mutations in primary B-AS compared to 82% of TP53 present in non-primary B-AS (p=0.02). Nine out of ten PI3KCA alterations were also seen in primary B-AS (P=0.0003) [11]. Despite sequencing techniques allowing the identification of recurrent somatic genetic abnormalities, the rarity of AS challenges our efforts to establish strong associations with the site of origin, etiology, and therapeutic implications.
Elaborate more on microenvironment and its genomics and signaling elements
As per reviewer request, additional background on microenvironment in STS is described in the Introduction line 108 to 118 and in methods line 132 to 139:
“Transcriptomic analysis is now available to estimate the relative abundance of immune and stromal cells within tumor samples. Using this technique, Petiprez et al. described a classification of soft tissue sarcomas based on their tumor microenvironment. In the SARC028 trial for the use of PDL-1 blockage for STS, they identified that an immune-rich microenvironment, particularly a B cell abundance, correlated with better response rate and improved PFS[22]. Interestingly, the overall TMB appeared similar across all classes of microenvironment phenotypes. In other histologies, microenvironment analysis also shows predictive capabilities for IO and other targeted therapies. For example, in renal cell carcinoma, gene expression signatures of angiogenesis, T-effector, and myeloid cells are predictive of PFS for IO alone or combined with anti-VEGF[23]. Whereas these methods can be applied similarly to patients with AS needs further investigation”.
“WTS was performed on 53 tumors and used for microenvironment cell population (MCP)-counter analysis as described by Becht et al. [24]. First, using transcriptomic markers (TM), estimating a cell population of interest from the sample is possible. TM are gene expression features expressed in one and only one cell population. The method generates an abundance score for CD3+ T cells, CD8+ T cells, cytotoxic lymphocytes, NK cells, B lymphocytes, monocyte lineage cells, endothelial cells, and fibroblasts[24]. Next, we identified subgroups based on tumor microenvironment profiles by hierarchical clustering of MCP-counter Z-scores [22]”.
Fig1 In Fig Legend explain why inflammatory cell are important for the sarcoma, genomics and biomarkers
We have added additional information on the importance of inflammatory cells for sarcoma.
Figure 1. B cell abundance and PDL-1 positivity is mostly seen in cases with grade 2 and 3 inflammatory infiltrate by H&E. Here, we illustrate the spectrum of the density of inflammation within angiosarcomas. a) grade 1 - <5% of cells are inflammatory cells. b) grade 2 - <30% of cells are inflammatory cells. c) grade 3 - >30% of cells are inflammatory cells.
Fig 2. In Figure legend elaborate more on dMMR/MSI, PD-L1, and Tumor Mutation Burden in the immunotherapy.
Figure 2. Immunotherapy response biomarkers vary according to the primary site. ** Head and neck angiosarcoma cases have a significantly higher predominance of TMB-High (>10 muts/MB) with p<0.0001. In addition, PDL-1 positivity is present among the different sites. AS cases rarely are dMMR/MSI- High.
Here, in the figure, we will not make any comments about these markers and immunotherapy response. Although the presence of these markers suggests IO activity, we cannot draw this conclusion from our cohort. Moreover, as in other tumors, classic biomarkers of IO response have predictive limitations.
Fig 3. In Fig Legend elaborate more on most important genes expressed
Figure 3. Figure 3. The genomic landscape of angiosarcoma shows a distinct pattern according to the primary site. a) Oncoprint for the entire cohort of 143 cases showing the most common alterations: TP53 (29%), MYC amp (23%), ARID1A (17%), POT1 (16%), and ATRX (13%). b) Genetic alterations vary by primary site. TP53 and POT1 are significantly higher in H-N AS; MYC amp is primarily seen in B-AS**p<0.0001
Fig 4. explain why for comparison was taken sarcoma and melanoma? Present heat map for sarcoma cells with explanation of most expressed genes important in clinics
Line 258 - 262: “Next, we compared the microenvironment of the AS cohort with a cohort of melanoma (n=1255). Of all tumors in humans, melanomas have one of the highest-burden of somatic genetic alterations[25]. Moreover, particularly in cutaneous melanoma, an extensive CD8+ T-cell infiltration has been described and associated with better prognosis and response to IO [26]".
The most common altered genes are shown in figure 3.
Fig 5 needs detailed explanation on gene expression
Fig. 5 The presence of a grade 2 or 3 inflammatory infiltrate observed on H&E microscopy correlates significantly with higher B cell abundance and PDL-1 expression. The expression of other immune checkpoint-related genes (CTLA4, TIM3, LAG3, PD1, PD-L1, PD-L2) follows that of the immune cells.
In the Discussion explain importance of sarcoma microenvironment genomics for the clinics. Explain in the details role of mutations found in sarcoma for the therapy and diagnostics.
Please see discussion line 331 to 358 for the implications of microenvironment in the clinics. We have mentioned above the therapeutic implications of the mutations. However, we decided not to expand in specific agents because strategies for targeting those genes are extensive and deserve a comprehensive analysis which is not the purpose of our manuscript.
“Here, we also describe a microenvironment with a high immune signature and abundance of B-cells is present in about 13% of the cases and evenly distributed among different primary sites. A signature of B-cell lineage abundance regardless of high or low CD8+ T cell infiltration appears to predict response to PD1 blockade and PFS in soft-tissue-sarcomas[22]. Interestingly, in this immune-high group, the cases had overall a low TMB. The contribution of each of these factors and the dynamic microenvironment changes to IO response are still unknown. In other solid tumors, distinct microenvironment characteristics show predictor capabilities to IO and other targeted therapies[19,20]. However, not a single phenotype across solid tumors has yielded similar prognostic and predictive capabilities. Whereas in AS, the phenotype with an abundance of B cells results in similar predictive capabilities of what has been seen for STS warrants further study. Additionally, the dynamics of the microenvironment upon treatment could potentially have better predictive capabilities. In melanoma, a highly immunogenic tumor where IO is active, INF-gamma driven infiltration of CD8+ lymphocytes upon treatment predicts responses[26,29]. In our cohort, we described a similar microenvironment to that of melanoma. We need to examine further the predictive capabilities of microenvironment analysis in AS to move forward to incorporating this method into clinical practice. WTS is commonly performed when profiling solid tumors; however, immune cell abundance and microenvironment cell counter results are not routinely reported to treating physicians. As more evidence of the predictive capabilities of this method comes out, we need to incorporate it into clinical decision-making as we move forward in tailoring effective therapies for rare tumors. Importantly, we reviewed the H&E slides of 138 of the patients (4 cases had no available H&E slide) and described the tumor inflammatory infiltrate. We saw that B-cell abundance by WTS and PDL-1 expression was associated with the presence of an inflammatory infiltrate of grade 2 or 3 as assessed by light microscopy. Thus, we could optimize this strategy by incorporating immunohistochemistry to compare the predictive phenotypes determined by the MCP-counter method. “

Round 3
Reviewer 2 Report
Manuscript is corrected as recommended
now can be published